# System identification and mechanical resonance frequency suppression for servo control used in single gimbal control moment gyroscope

Yue Yu[1,2,3], Lu Dai[1], Maosheng Chen[1]*, Lingbo Kong[1], Chaoqun Wang[1], Gengyao Li[1], Zhipeng Xue[1,2]

1 Chang Guang Satellite Technology Co., LTD, Changchun, China, 2 Changchun Institute of Optics, Fine Mechanics and Physics, Chinese Academy of Sciences, Changchun, China, 3 University of Chinese Academy of Sciences, Beijing, China

* chenms0911@aliyun.com

## Abstract

Effective identification of the control model is one of the key aspects in improving the performance of the single gimbal control moment gyroscope (SGCMG) servo system. The accuracy and stability of the servo system can be improved by studying system identification and mechanical resonance frequency. In this study, firstly, the SGCMG gimbal servo system was simplified to a two-mass block model. The theoretical mathematical model of the system's transfer function and mechanical resonance frequency was derived. Secondly, this paper studied the effective suppression method for mechanical resonance. Thirdly, the mathematical model of the orthogonal correlation analysis method was deduced for system identification. Then, an experimental platform was investigated to obtain the frequency characteristic curve and the transfer function. Finally, the frequency characteristic curve obtained using the transfer function model was plotted and compared with the frequency characteristic curve obtained experimentally. Our results indicate that the orthogonal correlation analysis has a high identification accuracy.

## 1. Introduction

The single gimbal control moment gyroscope (SGCMG)has the advantages of a large output torque, simple structure, frameless locking, and simple control, because of which it is widely used in spacecraft. Its application in spacecraft attitude control has always been an active area of research [1–3]. The rotor used in the SGCMG has a constant angular momentum. Therefore, the control accuracy mainly depends on the accuracy of the gimbal servo control system. The need to improve the accuracy of servo control system has led to renewed interest in the development of servo control algorithms in recent studies, including the closed-loop I/f control scheme [4], the sliding mode control method using a new approach combined with an iterative learning controller and extended state observer [5], and torque ripple minimization by modulation of the phase current [6].

**Data Availability Statement:** "Yes - all data are fully available without restriction" All relevant data

are within the paper and its Supporting Information files.

**Funding:** This work was supported in part by the Jilin Province Scientific and Technological Development Program, grant number 20180201111GX and 20210201114GX. The funders had no role in study design, data collection and analysis, decision to publish, or preparation of the manuscript.

**Competing interests:** The authors have declared that no competing interests exist.

The motor selected for the SGCMG servo system is a permanent magnet synchronous motor (PMSM) [7,8], which is a complex nonlinear system with strong coupling and multivariable characteristics. Therefore, it is necessary to establish an accurate mathematical model before studying the servo control algorithms. There are two methods to establish a mathematical model, one is by theoretical analysis [9–11], the other is by system identification through experiment. System identification obtains the dynamic characteristics of the system through its input and outpu to establish the mathematical model of the system. One of the most common problems in servo motion control systems is mechanical resonance [12,13]. Mechanical resonance is caused by the flexible connection between two or more components during mechanical transmission. Among them, the most critical part is caused by the flexible connection between the motor and the load. Mechanical resonance affects the control accuracy of a precision servo system and the mechanical resonance frequency determines the upper limit of the system bandwidth. When the mechanical resonance frequency is low and the system bandwidth requirement is high, it may cause the entire servo system to resonate and make the system unable to operate normally. In operation, strong resonance can even lead to bearing fracture and damage to the entire servo system [14]. Therefore, it is of considerable importance to identify control system models that can predict the mechanical resonance frequency. Kang et al. [15] indicated mechanical resonance often degrades the performance of the servo system and have employed shifted discrete Fourier translations to determine the properties of the mechanical resonance. Chen et al. [16] used a binary multifrequency excitation test signal to perform system identification. Yang et al. [17] explained the cause of mechanical resonance and have presented a tracking scheme using the velocity error and bandpass filters to track the mechanical resonance frequency.

In this study, the mechanical resonance frequency of the servo control was identified using orthogonal correlation analysis and a method for suppressing the mechanical resonance frequency was developed. Owing to the relative complexity of the PMSM and the difficulty in establishing an accurate system model through theoretical derivation and simulation, identifying the precise system transfer function is a prerequisite for accurate system control. The literature includes a strong body of work on the identification of the PMSM system model. Previous studies [18–22] identified the parameters of PMSM and then constructed the system model through theoretical model derivations. However, some simplifications and trade-offs are often made in the theoretical derivation process, and the real system is subjected to many external sources of interference. Therefore, theoretical model derivations would always be prone to a certain degree of error. Consequently, the determination of an accurate method of system model identification is still an active research subject in various fields of industrial research. An identification method was devised using the iterative process of the linearized and weighted total least-squares method [23]. Wen et al. [24] used a pseudo-random binary sequence as the input signal, acquired the impulse response based on correlation analysis, and derived the state space model from the impulse response through singular value decomposition and the frequency-domain model, obtaining the explicit values of variable parameters by parameter fitting. Ishak et al. [25] studied system identification using a fractional-order model because real systems and dynamical processes display fractional-orderbehavior.

Previous studies on system identification have not considered the identification accuracy and suppression of the mechanical resonance frequency. In this study, we conducted a system identification experiment to establish an accurate system model. Simultaneously, we developed a method for suppressing the mechanical resonance frequency. Our experiment verified the effectiveness of the mechanical resonance frequency suppression method and the identification algorithm. We obtained the amplitude–frequency and phase–frequency characteristics of the system through the orthogonal correlation analysis method. Further, we obtained the

transfer function and mechanical resonance frequency of the entire system by fitting the frequency characteristics.

The contribution of this study is the development of a system identification method that considers the mechanical resonance frequency and the system model. Moreover, methods for suppressing the mechanical resonance frequency were developed. In addition, we demonstrated that the orthogonal correlation analysis has a high identification accuracy.

The remainder of this paper is organized as follows. Section 2 describes the model of the SGCMG servo system and its mechanical resonance; it also details the method of mechanical resonance frequency suppression. Section 3 derives the orthogonal correlation analysis method for system identification. Section 4 describes the principle and the elements of the system identification experiment. Section 5 presents detailed results for the mechanical resonance frequency and the servo system model identification. Section 6 concludes the study.

## 2. Modeling and mechanical resonance frequency suppression of the SGCMG servo system

### 2.1 Modeling of the SGCMG servo system

The relationship between the motor assembly and the load of the SGCMG system is illustrated in Fig 1, where the shaft assembly, flywheel connector, and flywheel as a load are driven by the motor assembly.

To facilitate the analysis of the flexible coupling relationship between the motor and the load, we recognize that it is, in principle, equivalent to a two-mass model. Fig 2 depicts the equivalent model diagram.

The command current $i_q$ receives the feedback current $i_q^*$ through the current controller, and the feedback current is multiplied by the motor torque constant to obtain the electromagnetic torque $T_E$. The electromagnetic torque directly drives the PMSM to generate the motor acceleration $A_M$. The motor acceleration is integrated with respect to time to obtain the motor speed $V_M$, and the motor speed is integrated with respect to time to obtain the motor position $\theta_M$. When the motor starts running, the two-mass spring deforms and tightens so that the load produces a torque proportional to the difference in position of the motor and the load. The load torque causes the load to accelerate, producing an acceleration $A_L$, the load acceleration is integrated to obtain the load speed $V_L$, and the load speed is integrated to obtain the load position $\theta_L$. The torque transmitted by the motor causes the load to rotate, and the torque transmitted by the load further restricts the rotation of the motor.

The transmission principle block diagram is depicted in Fig 3. The stiffness torque $K_s^*(\theta_M - \theta_L)$ transmitted by the stiffness coefficient $K_S$ applies a negative torque to the motor and a positive torque to the load based on the position difference. The transmission of a damping torque is based on the velocity difference, and the rest of the transmission principle is the same as that for the stiffness torque.

According to the transfer function block diagram of the two-mass model of the servo system shown in Fig 3, the following matrix equation can be obtained:

$$\begin{bmatrix} T_E \\ 0 \end{bmatrix} = \begin{bmatrix} J_M s^2 + sK_{CV} + K_S & -sK_{CV} - K_S \\ -sK_{CV} - K_S & J_L s^2 + sK_{CV} + K_S \end{bmatrix} \begin{bmatrix} \theta_M \\ \theta_L \end{bmatrix} \tag{1}$$

where $T_E$ is the electromagnetic torque output by the motor, $J_M$ is the moment of inertia of the motor, $J_L$ is the moment of inertia of the load, $K_S$ is the stiffness coefficient of the system, $K_{CV}$ is the damping coefficient of the system, $\theta_M$ is the angular position of the motor output, and $\theta_L$ is the angular position of the load output.

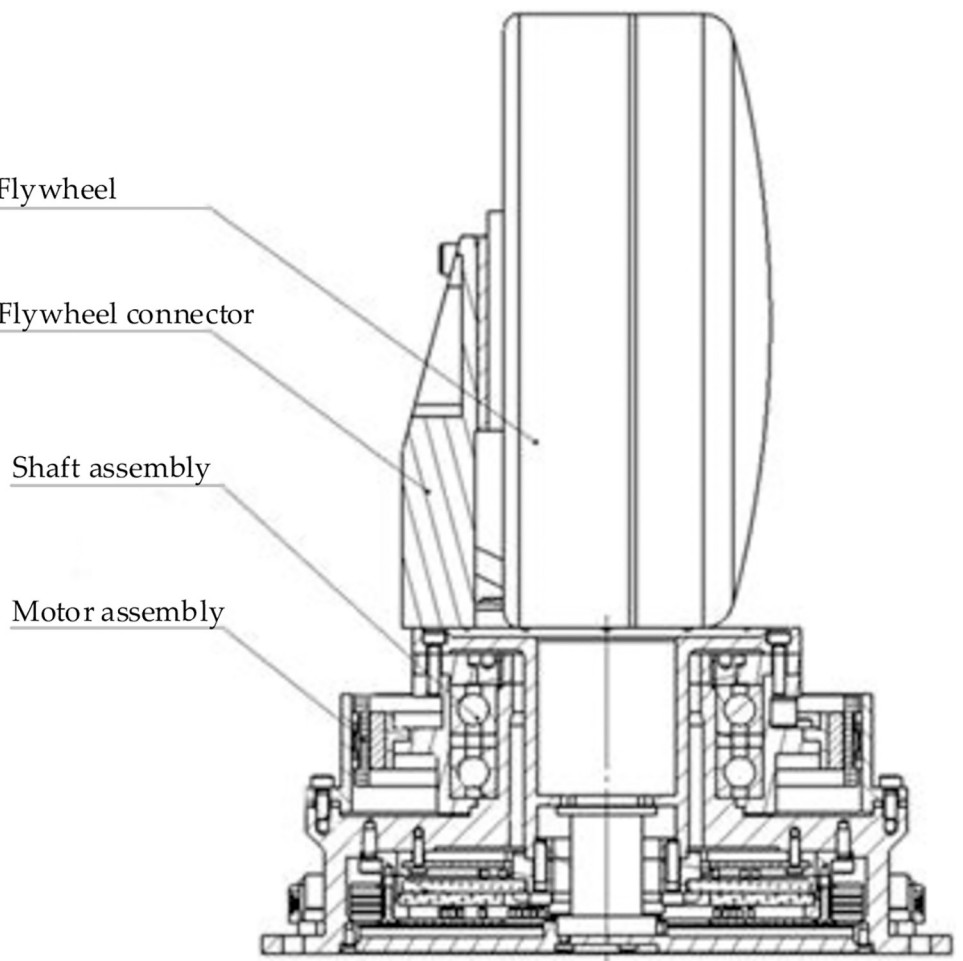

**Fig 1. SGCMG system structure diagram.**

According to Eq (1), it is easy to deduce the transfer function between the electromagnetic torque $T_E$ of the motor and the output angular velocity $V_M$ of the motor as shown in Eq (2):

$$\frac{V_M(s)}{T_E(s)} = \frac{1}{(J_M + J_L)s}\left[\frac{J_L s^2 + K_{CV}s + K_S}{\frac{J_M J_L}{J_M + J_L}s^2 + K_{CV}s + K_S}\right] \tag{2}$$

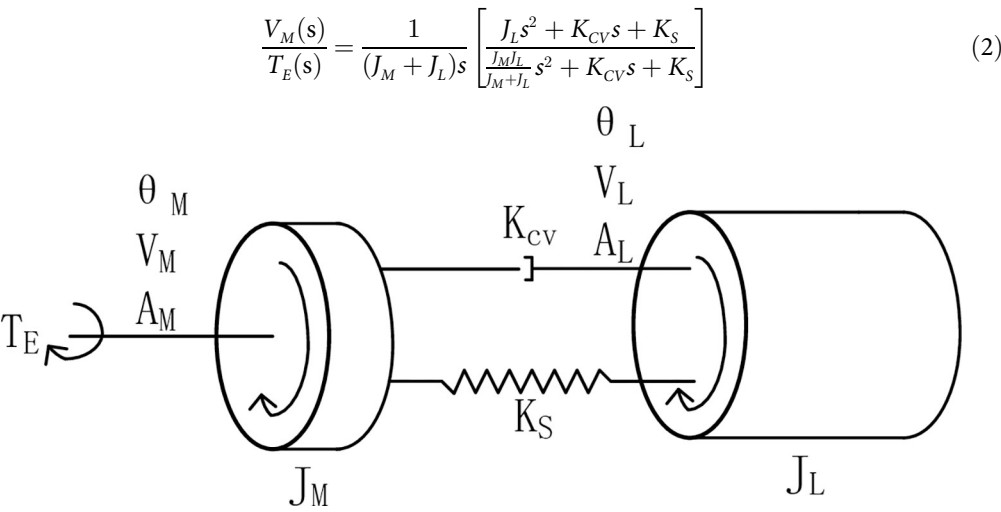

**Fig 2. Servo system two-mass model.**

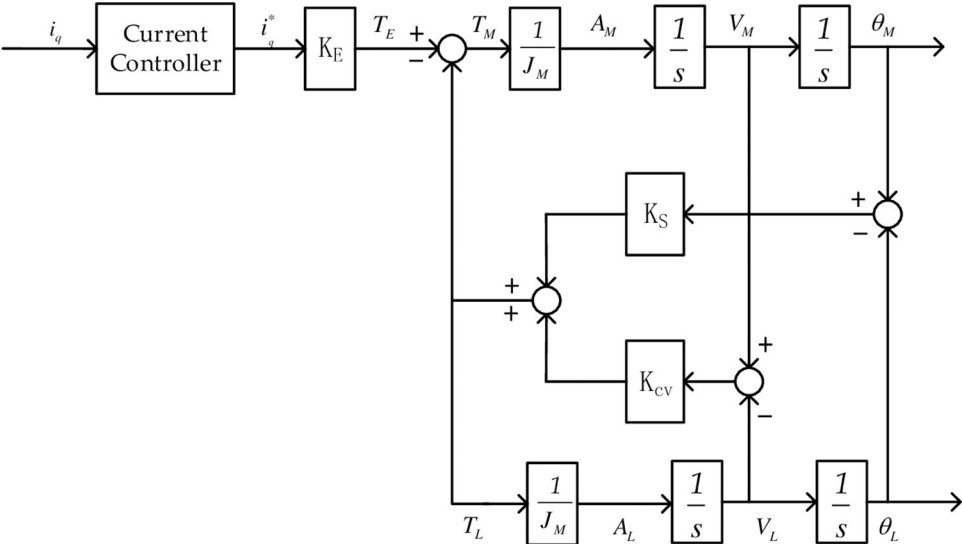

**Fig 3. Servo system two-mass model transfer function block diagram.**

## 2.2 Mechanical resonance frequency suppression

In order to facilitate the introduction of the mechanical resonance frequency, the frequency characteristic curve containing the mechanical resonance frequency is shown in Fig 4. The gain in the amplitude–frequency characteristic of the valley frequency is called the anti-resonance frequency ($f_{AR}$). The relationship between $f_{AR}$, the stiffness coefficient, and the moment of inertia of the load is shown in Eq (3). According to Eq (3), the anti-resonant frequency is the natural oscillation frequency of the load and the spring, which is independent of the motor, but the system is difficult to operate under the anti-resonant frequency condition as all the energy input into the motor will flow quickly to the load. Although the motor is near to a calm state at the anti-resonant frequency, the load may be oscillating at a higher intensity at that time.

$$f_{AR} = \sqrt{\frac{K_S}{J_L}} rad/s \tag{3}$$

The peak frequency of the gain at the amplitude–frequency characteristic is called the resonant frequency ($f_R$). The expression of the resonant frequency is shown in Eq (4). At this resonant frequency, the motor and the load offer almost no obstacle to the movement. As the total inertia is small, the loop gain becomes large.

$$f_R = \sqrt{\frac{K_S(J_M + J_L)}{J_M J_L}} rad/s \tag{4}$$

As the motor and the load are in free oscillation at the anti-resonance frequency, the focus is usually on the anti-resonance frequency rather than on the mechanical resonance frequency. The mechanical resonance frequency will lead to extreme instability of the motor and the load, imposing much greater difficulty on the controller design. In this study, the occurrence of the mechanical resonance frequency is suppressed by taking the following three measures:

1. Increasing the ratio of the motor's rotational inertia to the load's rotational inertia.

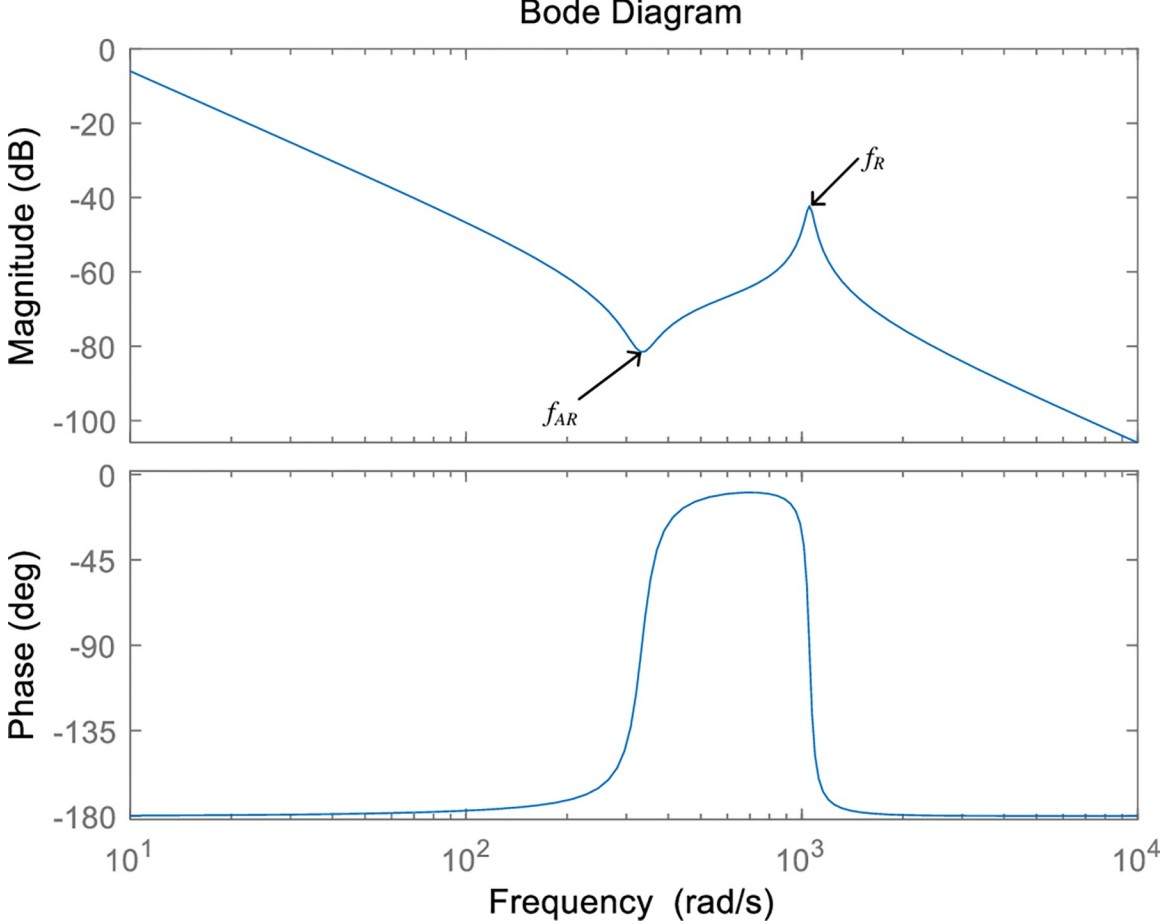

**Fig 4. Frequency characteristic curve including the mechanical resonance frequency.**

Increasing the ratio between the motor's rotational inertia and the load's rotational inertia is the most reliable method for improving the resonance problem. This is because a larger ratio of the motor's rotational inertia to the load's rotational inertia will result in less flexure of the system, and the prevalent resonance problems are caused by the flexure between the motor and the load. An increase in the ratio between the motor's rotational inertia and the load's rotational inertia can be achieved by either increasing the rotational inertia of the motor or decreasing the rotational inertia of the load. Increasing the motor's rotational inertia will also increase the total inertia of the system. Assuming that the motor's rotational inertia is increased by 25%, the available acceleration will be reduced by 25%; consequently, the torque would need to be increased by 25% to ensure the same acceleration as before. Increasing the motor's rotational inertia will increase the cost of both the driver and the motor, which is contrary to the development goals of commercial satellites of high precision, low cost, low weight, and low power consumption. Therefore, reducing the rotational inertia of the load is the best way to increase the ratio of the motor's rotational inertia to the load's rotational inertia. As rotational inertia is a physical quantity that reflects the mass distribution of a rigid body, the rotational inertia of a load can be reduced by either reducing the mass of the load, or changing the dimensions of the load, or both. The magnitudes of the motor's rotational inertia and the load's rotational inertia of the SGCMG gimbal servo system designed in this study are shown in Table 1.

**Table 1. Rotational inertia of the SGCMG gimbal servo system.**

| Parameter | $J_M$ | $J_L$ |
|---|---|---|
| Value | 44340000 g·mm$^2$ | 10058854.28 g·mm$^2$ |

2. Increasing the stiffness of the system.

Increasing the stiffness of the SGCMG rotating mechanism is another important measure to prevent the occurrence of mechanical resonance. The SGCMG rotating mechanism is shown in Fig 5. It is designed to increase the stiffness by including the following features:

a. The direct drive of the motor is used to avoid the introduction of additional flexible links.

b. A compact structure design is adopted to shorten the length of the transmission shaft.

c. Subject to the weight and size constraints, the diameter of the transmission shaft is increased as much as possible.

d. The rotating mechanism structures are designed using materials with high specific stiffness.

e. An axial preload is applied on the bearing to improve the supporting stiffness, rotation accuracy, and stability of the bearing unit.

Following the modal analysis of the entire machine construction of the SGCMG system, the frequency nephogram of each order is shown in Fig 6, and the natural frequency of each order is shown in Table 2. According to the modal analysis, the system possesses sufficient stiffness and avoids being detrimentally affected by the flywheel rotation frequency or the natural frequency of the gimbal structure; effectively, the design avoids the occurrence of resonance problems.

3. Designing filters.

The resonance problem can be further attenuated in the SGCMG gimbal servo control system by designing filters, which are placed before the current loop and used to attenuate the gain variation caused by the flexibility between the motor and the load. The filters can be low-pass filters or hysteresis filters; adding filters can reduce the gain around the resonance frequency. The transfer function of the low-pass filter is shown in Eq (5). When the frequency of the low-pass filter is increased to the turning frequency $\omega$, the attenuation becomes increasingly severe, and the low-pass filter causes a relatively large phase lag problem, which is equivalent to reducing the phase margin to some extent. The transfer function of the hysteresis filter is shown in Eq (6), where $\omega_2 < \omega_1$, which has the maximum attenuation $\omega_2/\omega_1$ at high frequencies. As the hysteresis filter has the advantage of a relatively small phase lag, a hysteresis filter is added before the current loop in this study.

$$G_{fl} = \frac{1}{s/\omega + 1} \tag{5}$$

$$G_{fd} = \frac{s/\omega_1 + 1}{s/\omega_2 + 1} \tag{6}$$

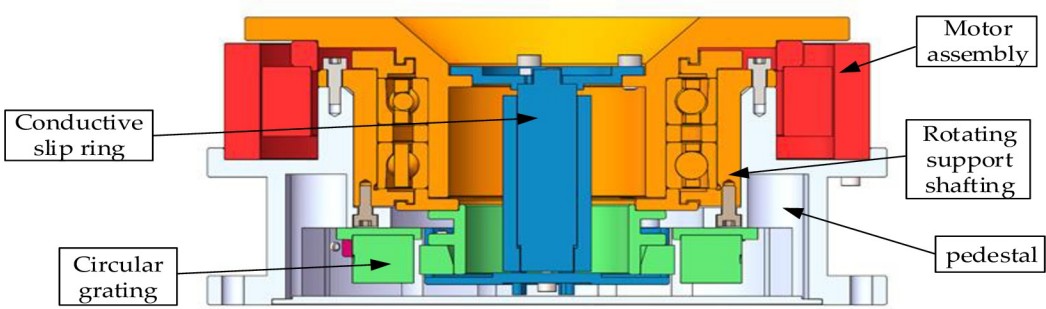

**Fig 5. Schematic diagram of the SGCMG rotating mechanism.**

## 3. Modeling the SGCMG servo system identification method

The SGCMG servo system identification process treats the SGCMG servo system as a black box [26]. A given input signal generates a specific output signal. The particular black-box model for a system is identified according to the characteristics of the input and output signals. According to the relevant identification algorithm, the frequency characteristic curve of the input and output signals is obtained, and then the transfer function of the black-box model of the system is identified. The system identification process is shown in Fig 7.

First, an input of the corresponding excitation signal is made to the system. The speed response signal calculated by the circular grating of the SGCMG servo system is recorded. Secondly, the appropriate identification algorithm is used to obtain the spectrum characteristic response curve of the system. Then, the transfer function of the system is obtained by curve fitting.

The input excitation signal can take the form of either a white noise signal or a sinusoidal frequency sweep signal. Because the ideal white noise signal is difficult to generate, it will lead to poor repeatability of the identification experiment if the white noise signal is chosen as the excitation signal. In addition, because the frequency components contained in white noise cannot be controlled, there will be no excitation signal at certain frequencies, resulting in large errors in the calculation. Therefore, we select a sinusoidal frequency sweep signal as the input excitation signal in this study. The input signal frequency should be selected to ensure that the lowest frequency is at most 50% of the first corner frequency of the servo system. It is simultaneously necessary to ensure that the amplitude of the input current signal is reasonable. If the input current amplitude is too large, it will lead to an inappropriately high speed of the SGCMG servo system, which may cause permanent damage to the SGCMG. If the input current amplitude is too small, it would appear (because of the servo system) to overcome the friction of the shaft system, resulting in the entire system having no output response. As for the sampling frequency of the input and output signals, according to the Shannon sampling theorem, the sampling frequency of the signal should not be less than twice the highest resonant frequency of each link of the servo system.

The servo system frequency characteristics can be classified according to the type of input signal: deterministic (which can be described by an analytic expression), random, or pseudo-random (which can be described by an analytic expression while approaching a random nature). The signal processing method can be either a frequency response test, a Fourier analysis, or a correlation analysis. The frequency response test requires the evaluation of multiple frequency points to obtain the frequency response characteristics of the system; the accuracy of the result obtained with this test is relatively high, and the use of orthogonal correlation analysis is highly effective in this test. Fourier analysis can identify the frequency response of a

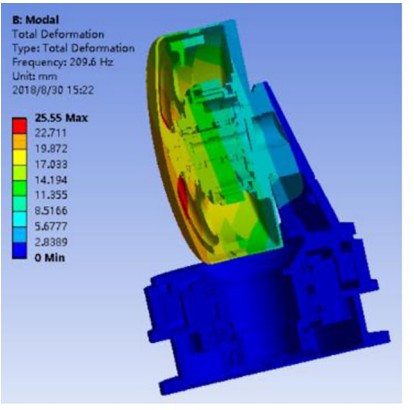

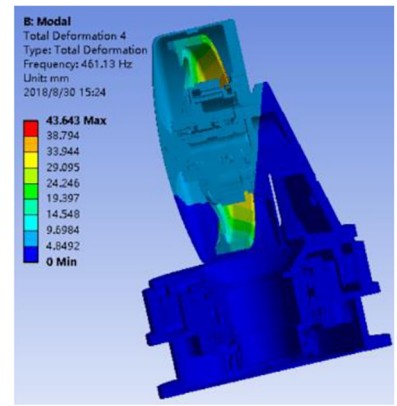

(a)    First-order

(b)    Second-order

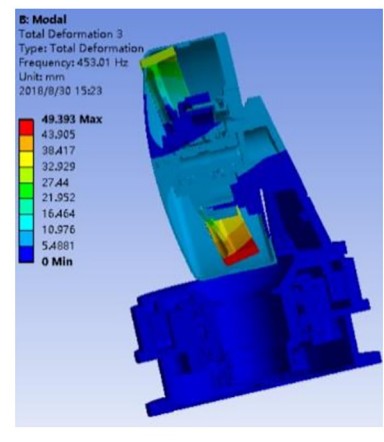

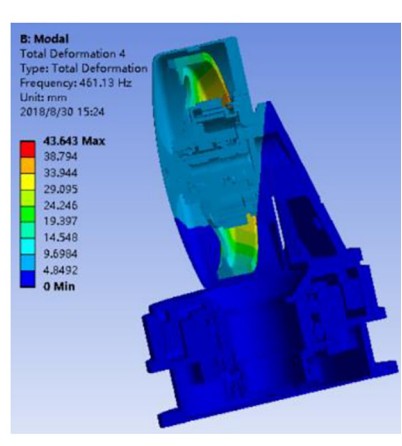

(c)    Third-order

(d)    Fourth-order

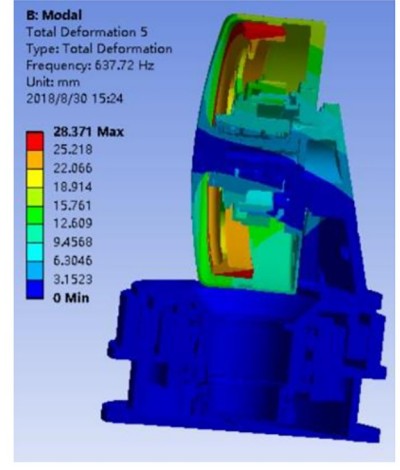

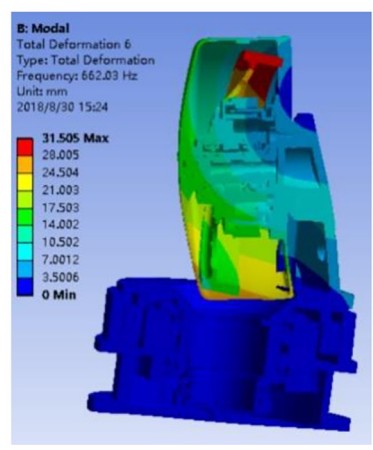

(e)    Fifth-order

(f)    Sixth-order

**Fig 6. Frequency nephogram of each order.**

**Table 2. Natural frequency of each order.**

| Order | Frequency (Hz) |
|---|---|
| First-order | 209.6 |
| Second-order | 226.21 |
| Third-order | 453.01 |
| Fourth-order | 461.13 |
| Fifth-order | 637.72 |
| Sixth-order | 662.03 |

linear system based on impulse and step responses; although this method is computationally less intensive, relatively simple to implement, and can be performed in a short time, it is only suitable for systems with a good signal-to-noise ratio. The correlation analysis method must be performed in the time domain, targeting linear systems that can be either continuous or discrete time signals, and it is equally applicable under poor signal-to-noise ratio conditions.

The input signal selected in this study is a deterministic signal, and the signal processing method is selected based on the frequency response test of the orthogonal correlation analysis method to obtain the amplitude–frequency and phase–frequency characteristic curves of the SGCMG servo system. We describe the following orthogonal correlation analysis for obtaining the frequency response processing method:

For a linear system, the autocorrelation function of the input signal is written as:

$$R_{uu}(\tau) = \lim_{T \to \infty} \frac{1}{T} \int_{-\frac{T}{2}}^{\frac{T}{2}} u(t)u(t+\tau)dt, \tag{7}$$

where $u(t)$ is the input response, $T$ is the sampling period, and $\tau$ is the time delay.

The cross-correlation function can be written as follows:

$$R_{uy}(\tau) = E\{u(t)y(t+\tau)\}$$

$$= \lim_{T \to \infty} \frac{1}{T} \int_{-\frac{T}{2}}^{\frac{1}{T}} u(t)y(t+\tau)dt$$

$$= \lim_{T \to \infty} \frac{1}{T} \int_{-\frac{T}{2}}^{\frac{1}{T}} u(t-\tau)y(t)dt \tag{8}$$

where $y(t)$ is the output response.

Through a convolution integral, the two correlation functions (7) and (8) can be combined to obtain:

$$R_{uy}(\tau) = \int_{0}^{\infty} g(t')R_{uu}(\tau - t')dt', \tag{9}$$

where $g(t')$ is the impulse response. The frequency response of the system can be determined

**Fig 7. System identification flowchart.**

by the impulse response, and the Fourier transform of the impulse response can be performed to obtain the frequency response $G(\omega)$.

For sinusoidal input, the signal can be expressed as:

$$u(t) = u_0 \sin\omega_0 t. \qquad (10)$$

In this equation, $u_0$ is the input signal amplitude, and $\omega_0$ is the input signal frequency. According to Eqs (7) and (10), the autocorrelation function can be written as:

$$R_{uu}(\tau) = \frac{2u_0^2}{T} \int_0^{\frac{T}{2}} \sin\omega_0 t \sin(\omega_0(t+\tau)) dt \\ = \frac{u_0^2}{2} \cos\omega_0 \tau \qquad (11)$$

The expression of the output response $y(t)$ can be given as:

$$y(t) = u_0 |G(\omega_0)| \sin(\omega_0 t - \varphi(\omega_0)), \qquad (12)$$

where $\varphi(\omega_0)$ represents the frequency response phase.

According to Eq (9), the cross-correlation function between the output response and the input signal is obtained as follows:

$$R_{uy}(\tau) = |G(\omega_0)| \frac{2u_0^2}{T} \int_0^{\frac{T}{2}} \sin\omega_0(t-\tau)\sin(\omega_0 t - \varphi(\omega_0)) dt \\ = |G(\omega_0)| \frac{u_0^2}{2} \cos(\omega_0 \tau - \varphi(\omega_0)) \qquad (13)$$

Eq (13) leads to the following:

$$|G(\omega_0)| \cos(\omega_0 \tau - \varphi(\omega_0)) = \frac{R_{uy}(\tau)}{\frac{u_0^2}{2}}. \qquad (14)$$

The real and imaginary parts of the frequency response can be estimated from Eq (14). When $\tau = 0$, the real part of the frequency response can be expressed as:

$$\mathrm{Re}\{G(\omega_0)\} = |G(\omega_0)| \cos(\varphi(\omega_0)) = \frac{R_{uy}(0)}{\frac{u_0^2}{2}}. \qquad (15)$$

When $\tau = \frac{\pi}{2\omega_0}$, the imaginary part of the frequency response can be expressed as:

$$\mathrm{Im}\{G(\omega_0)\} = |G(\omega_0)| \sin(\varphi(\omega_0)) = \frac{R_{uy}\left(\frac{\pi}{2\omega_0}\right)}{\frac{u_0^2}{2}}. \qquad (16)$$

According to Eq (8), by multiplying and integrating the input signal and the output signal of the system, the cross-correlation function when $\tau = 0$ can be calculated as:

$$R_{uy}(0) = \frac{u_0^2}{2}\text{Re}\{G(\omega_0)\} = \frac{u_0}{nT}\int_0^{nT} \text{y(t)}\sin\omega_0 t dt, \qquad (17)$$

where $n$ is the number of measurement cycles.

Similarly, the cross-correlation function when $\tau = \frac{\pi}{2\omega_0}$ can be calculated as:

$$R_{uy}(\frac{\pi}{2\omega_0}) = \frac{u_0^2}{2}\text{Im}\{G(\omega_0)\} = -\frac{u_0}{nT}\int_0^{nT} \text{y(t)}\cos\omega_0 t dt, \qquad (18)$$

where the phase shift of the velocity measurement signal is $\pi/2$ so that the sine signal becomes a cosine signal with the same frequency.

According to the following equation, the amplitude and phase relationships of the system's frequency response can be determined as follows:

$$|G(\omega_0)| = \sqrt{\text{Re}^2\{G(\omega_0)\} + \text{Im}^2\{G(\omega_0)\}}, \qquad (19)$$

$$\varphi(\omega_0) = \arctan\frac{\text{Im}\{G(\omega_0)\}}{\text{Re}\{G(\omega_0)\}}. \qquad (20)$$

## 4. SGCMG servo system identification experiment

### 4.1 Principle

The system identification principle diagram is shown in Fig 8. The system identification experiment adopts the vector control strategy of $i_d = 0$. The computer provides an input signal to the SGCMG gimbal servo system. According to whether the current loop is a closed loop or not, there are two waysfor signal input:

1. If the current loop is open, a voltage signal is provided as an input to the system. The object of identification includes the motor part, the coupling part of the motor and the load, and the drive amplifier part.

2. If the current loop is closed, a current signal is provided as an input to the system. The object of identification includes the current loop controller, the motor part, the coupling part of the motor and the load, and the drive amplifier part.

As the current loop is the innermost loop of the SGCMG servo control system, the current loop is part of the open-loop model of the speed loop. Therefore, in this study, we choose the second scheme, which uses the closed loop of the current loop. Using the programmable characteristics of the controller, the current signal is input to the servo system at point I to drive the SGCMG gimbal servo system to rotate. Simultaneously, the frequency is increased from low to high. The system synchronously records the velocity response signal of the SGCMG servo system calculated by the circular grating at point O. The sampling can be stopped after the set sampling period is completed, and the data is stored in the computer.

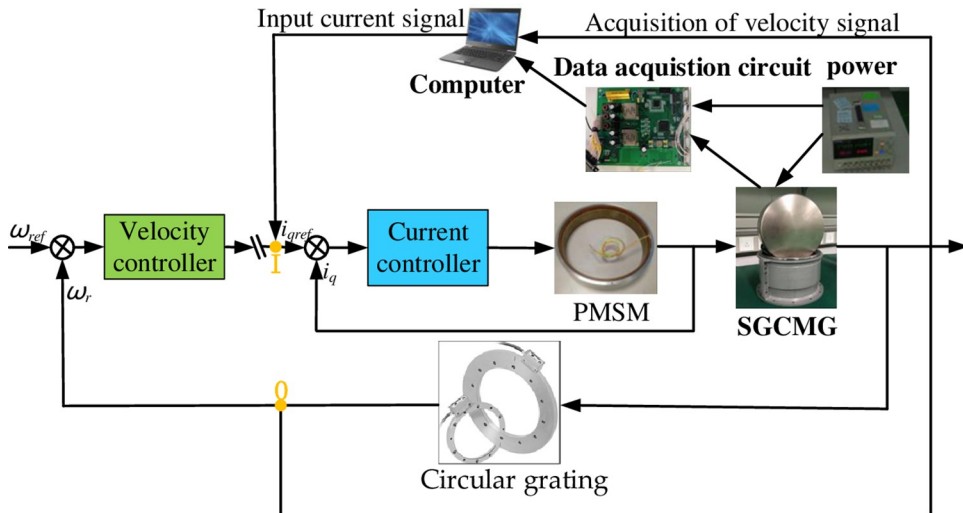

**Fig 8. System identification principle diagram.**

## 4.2 Execution of servo system identification experiment

The experimental system includes the SGCMG system, a data acquisition circuit, a power supply, and a computer, as shown in Fig 9. The data acquisition circuit is used for transmitting the control signal and driving the motor rotation. The power supply provides a voltage of 28 V to the PMSM and a voltage of 5 V to the digital control chip. The computer is used to pass control instructions to the entire system and to write the control flow code. The software for data recording and processing is developed in-house. The current signal are provided as inputs to the SGCMG system, and the velocity signal, which is obtained through the circular grating differential, is transmitted by the controller area network (CAN) to the computer for recording.

The main flow of the frequency characteristic identification experiment for the SGCMG gimbal servo system is specified as follows:

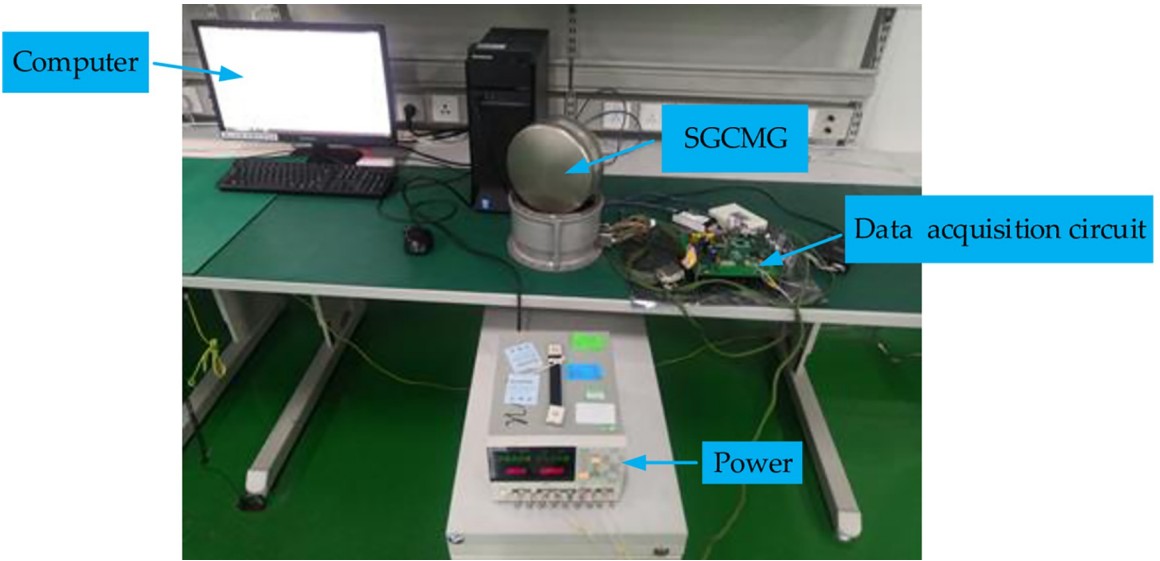

**Fig 9. System identification experiment.**

**Table 3. The parameters of the sinusoidal frequency sweep signal.**

| Parameter | A | $f_0$ | $f_T$ | T | n |
|---|---|---|---|---|---|
| Value | 0.08 A | 1.2 HZ | 100 HZ | 30 s | 3 |

1. Time-domain test: a current signal with appropriate frequency and amplitude is selected as the input signal of the system to drive the SGCMG gimbal to overcome the friction torque rotation, and the angular velocity response signal is collected and stored in real time.

2. Frequency-domain estimation: the orthogonal correlation analysis method is used to perform the calculation of the input and output time-domain signals collected in the system identification experiment. The frequency characteristic curves containing the amplitude–frequency and the phase–frequency characteristics are obtained using the previouslyde-scribed method.

3. Fitting transfer function: the transfer function model of the SGCMG gimbal servo system is obtained by the curve fitting method, in accordance with the frequency characteristic curve, and the frequency characteristic curve of the fitting transfer function is plotted and compared with that obtained by experiment.

An appropriate input signal needs to be designed as an excitation signal to ensure the experiment is appropriate and rigorous. Some scholars take a white noise [27] signal with an appropriate frequency band as the input excitation which can give a balanced excitation of each frequency point of the tested system and allows the dynamic frequency characteristics of the tested system to be obtained. However, a white noise signal with random characteristics does not exist in reality, and number of the frequency bands of white noise that can be realized physically are limited, so the repeatability of using white noise signal as input excitation signal for system identification is relatively poor. The sinusoidal frequency sweep method is a fast and effective measurement method of frequency characteristics. Compared with the white noise method, the biggest advantage is that the repeatability is better. In this paper, a sinusoidal frequency sweep signal is used as the input excitation signal of the system to test the frequency characteristics of SGCMG gimbal servo system. The sinusoidal frequency sweep signal is programmed in the controller according to Formulas (21)–(23).

$$I(t) = A\sin(2\pi \times u(t)), \tag{21}$$

$$u(t) = f_0(1 + st^n)\, t, \tag{22}$$

$$s = \frac{f_T/f_0 - 1}{(n + 1)\, T^n}. \tag{23}$$

Where $I(t)$ is the sinusoidal frequency sweep signal, $A$ is the amplitude, $f_0$ and $f_T$ are the starting point and end point of sinusoidal frequency sweep signal respectively, $T$ is the period required for the servo system to complete the frequency sweep, and $n$ is the polynomial order of the transfer function to identify the controlled object.

The parameters of the sinusoidal frequency sweep signal used in this paper are shown in Table 3 below.

## 5. Results and discussion

In order to verify the correctness of the experimental platform, a single frequency sinusoidal current signal is used as the input to the SGCMG gimbal servo system. The frequency and

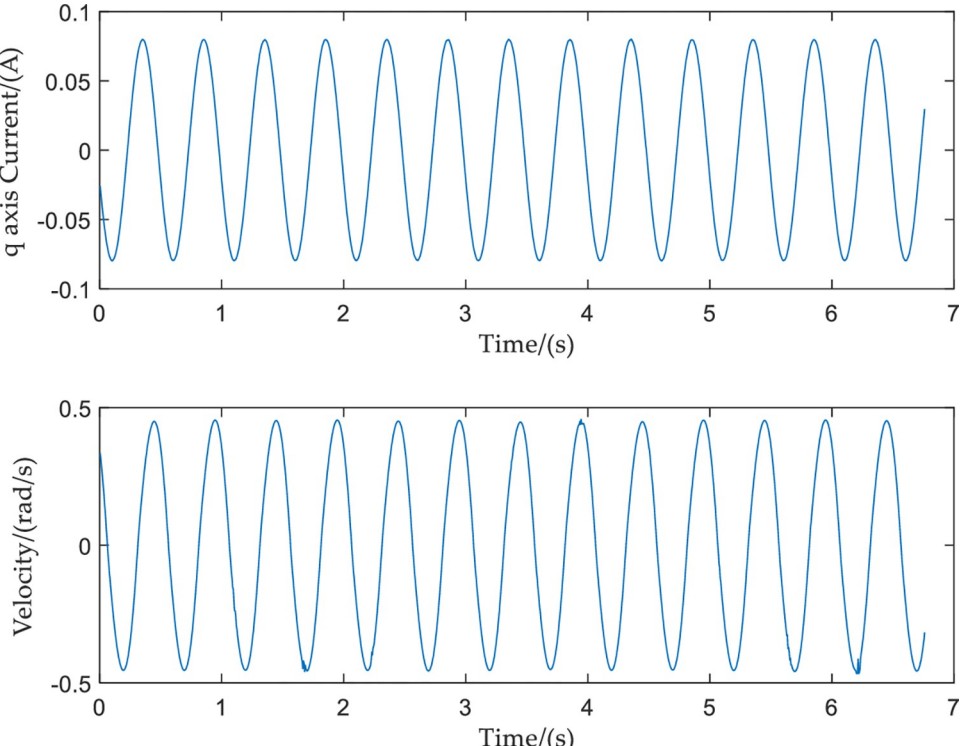

**Fig 10. Single frequency input and output curves.**

amplitude of the input signal, respectively, are 2 Hz and 0.08 A. The frequency and amplitude of the output velocity response signal, respectively, are 2 Hz and 0.4573 rad/s. The input and output curves are shown in Fig 10.

Because the above experiment only uses one frequency signal as the input, only one amplitude–frequency characteristic and one phase–frequency characteristic can be obtained. The frequency characteristic curve of the system obtained by repeatedly collecting a single frequency point in this way is not only time-consuming and labor-intensive, but the result obtained is also not accurate. Therefore, the input current is changed to a sinusoidal frequency sweep current signal. The frequency and amplitude of the input signal, respectively, are 1.2–100 Hz and 0.08 A. The input sine sweep signal and the output speed response signal have the same sampling frequency of 200 Hz. The resulting curves of the input sine sweep current signal, and the output velocity response, are shown in Fig 11.

Using the method of orthogonal correlation analyze to analyse the input sine sweep current signal and the output response signal, the frequency characteristic curve of the SGCMG servo system can be obtained, as shown in Fig 12. In addition to the orthogonal correlation analysis method used in this paper, the identification algorithm often used by other scholars is the Fourier transform [28–33]. In order to prove the superiority of the identification algorithm proposed in this paper, the frequency characteristic curve of the SGCMG gimbal servo system was obtained by the Fourier transform at the same time. The comparison of the frequency characteristic curve generated by the two methods is shown in Fig 13. Evidently, the noise on the frequency response characteristic curve calculated by the orthogonal correlation analysis method is smaller. As the computer is limited by measurement time, the transform can only be carried out in a limited range; this reduces the accuracy and produces the noise on the system output signal. The orthogonal correlation analysis method multiplies the input excitation signal and

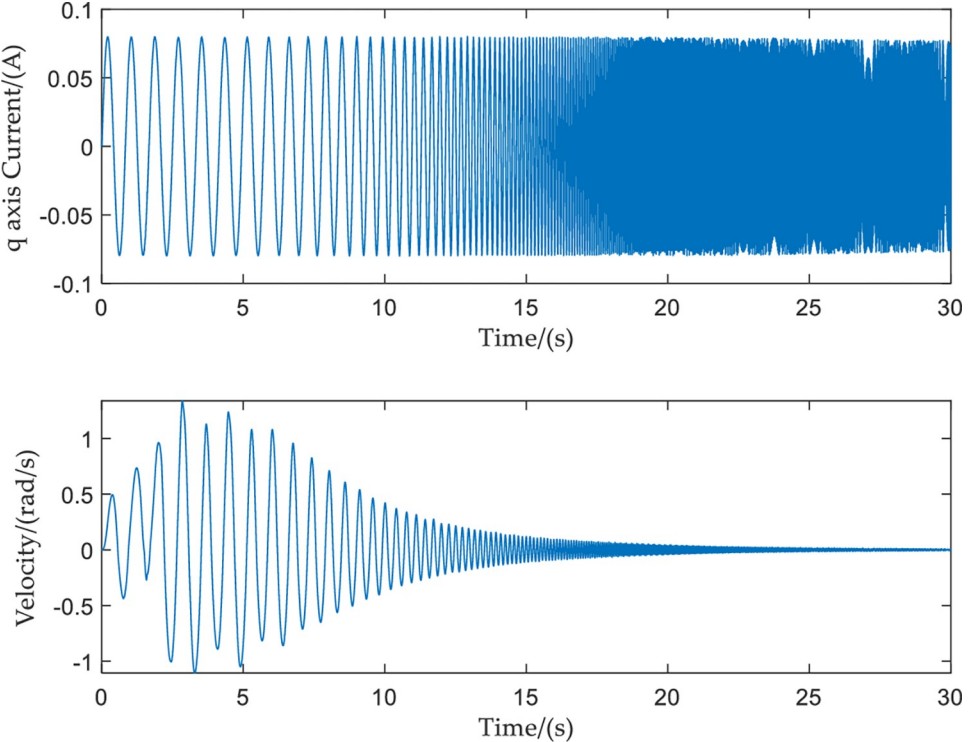

**Fig 11. Sinusoidal frequency sweep signal input and output curves.**

output response of the system to calculate the average value, so it can effectively suppress the interference harmonics and noise.

Fig 12 indicates that the SGCMG gimbal servo system designed in this study can effectively prevent mechanical resonance by increasing the ratio of the motor rotational inertia to the load rotational inertia, increasing the stiffness of the system, and designing a filter. The system transfer function *G(s)* can be obtained by fitting the frequency characteristic curve of the SGCMG servo system, as shown in Eq (24). The result shown in Fig 13 displays the comparison between the frequency characteristics measured in the experiments and those calculated by the theoretical fitting. Fig 14 indicates that the amplitude–frequency and phase–frequency characteristics of the transfer function identified agree well with the measured curve. Fig 15 is the error curve between the measured frequency characteristic and the frequency characteristic of the identified transfer function. It is evident from this curve that the absolute errors in the amplitude–frequency and phase–frequency characteristics are less than 0.5 dB and 1˚, respectively, in the common used frequency band.

$$G(s) = \frac{V_M(\text{s})}{T_E(\text{s})} = \frac{320s^2 + 3203000\text{s} + 32000000}{0.17\text{s}^3 + 854\text{s}^2 + 2000\text{s}}. \tag{24}$$

## 6. Conclusions

This paper mainly focuses on the study of mechanical resonance and transfer function using the system identification method. First, the SGCMG gimbal servo system was modelled as the equivalent of a two-mass block model, and the transfer function between the electromagnetic torque and the output angular velocity of the gimbal servo system motor was studied. Second, the typical mechanical resonance frequency in the general servo system was considered. By

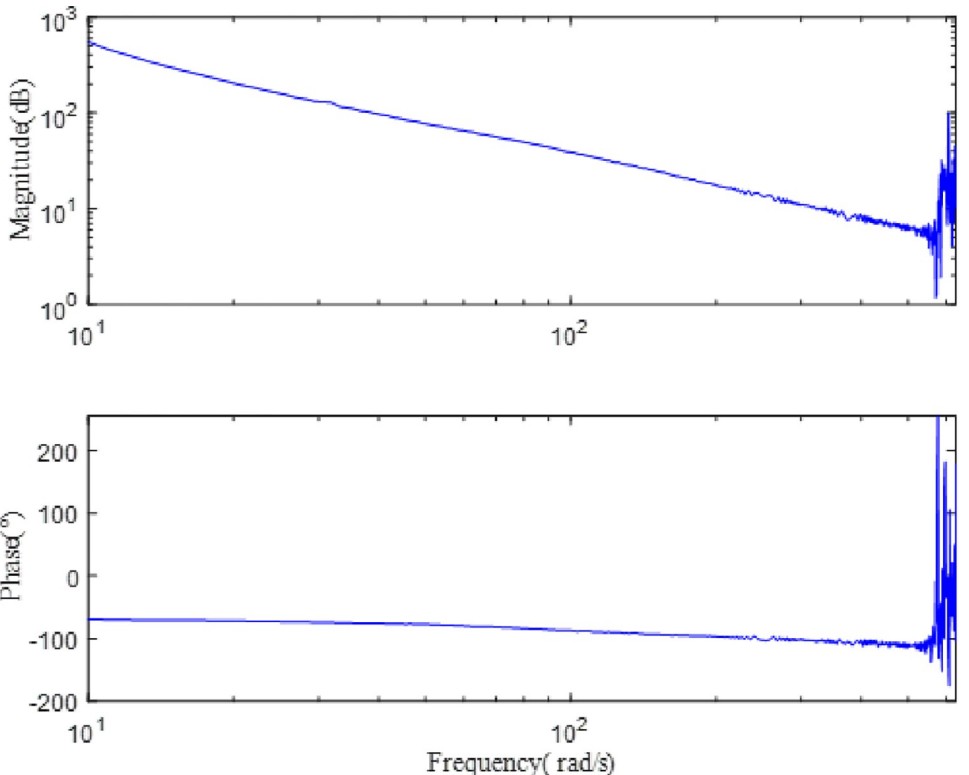

**Fig 12. The frequency characteristic curve of the SGCMG servo system.**

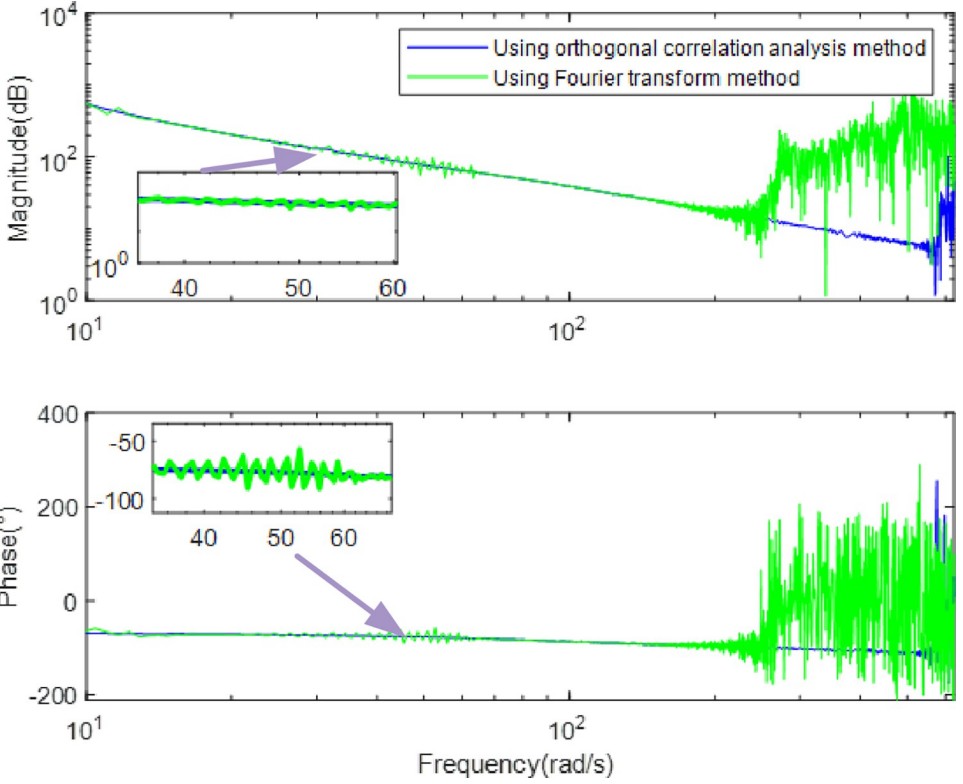

**Fig 13. Comparison of frequency characteristic curves obtained by orthogonal correlation analysis and Fourier transform method.**

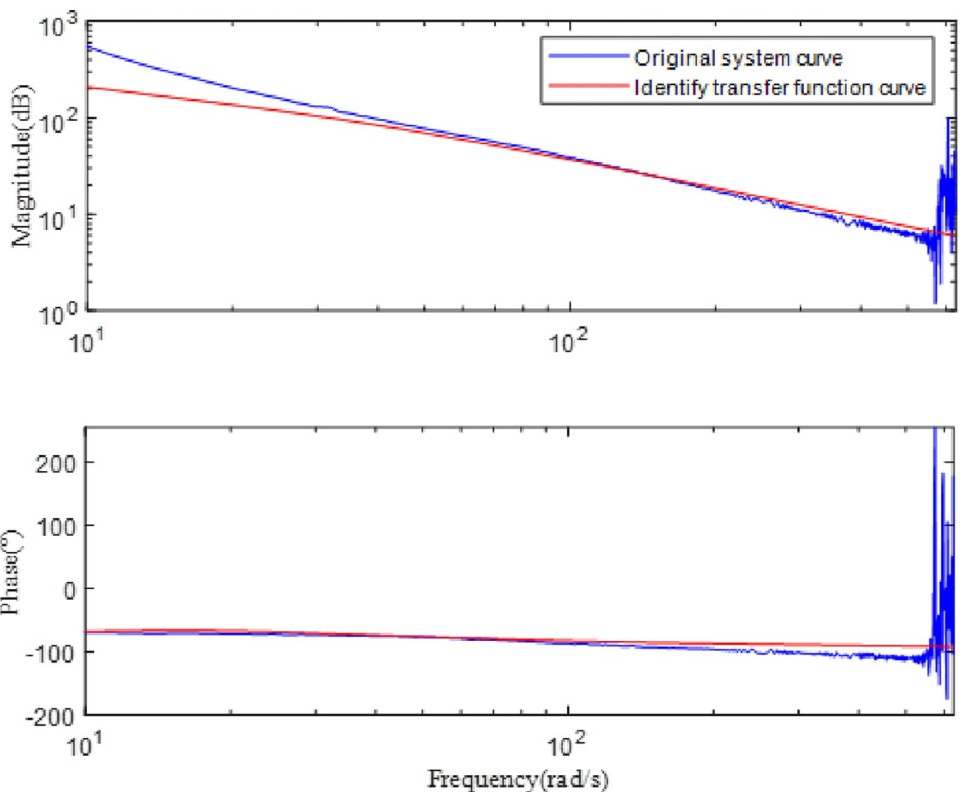

**Fig 14. Frequency characteristic curves of acquisition and identification.**

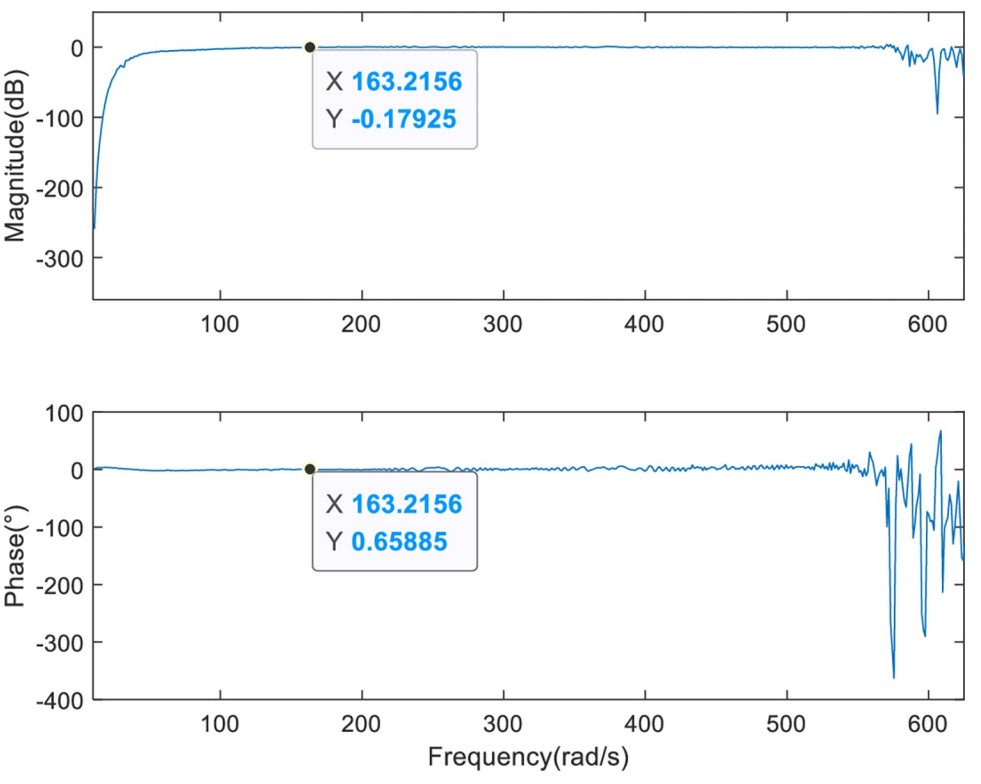

**Fig 15. Identification errors.**

starting from the underlying cause of the mechanical resonance frequency, methods of preventing the mechanical resonance were proposed. Finally, an input sinusoidal current sweep signal was used to test the frequency characteristics of the system. The input sinusoidal current sweep signal and the output velocity response signal were analyzed and processed by the orthogonal correlation analysis method. The frequency characteristic curves of the system, including the amplitude–frequency and the phase–frequency characteristics, were obtained. According to the frequency characteristic curve, it can be observed that the SGCMG designed in this study has no mechanical resonance problem. The novelty of this research is in proposing the application of orthogonal correlation analysis to servo system identification and comparing the identification results with the widely used Fourier transform method. The results show that the frequency characteristic curve identified by the orthogonal correlation analysis method has less noise singal than that obtained by Fourier transform. In addition, this paper also proposes the following options to prevent mechanical resonance: increase the ratio of the motor rotational inertia to the load rotational inertia, increase the stiffness of the system, and design a filter; these steps can improve the stability of the servo system.

## Supporting information

**S1 Data.**
(RAR)

## Acknowledgments

We thank the 13th Research Institute of the 9th Research Institute of China Aerospace Science and Technology Corporation for their motor.

## Author Contributions

**Conceptualization:** Yue Yu, Lu Dai, Maosheng Chen, Chaoqun Wang.

**Data curation:** Yue Yu, Lu Dai.

**Project administration:** Maosheng Chen.

**Resources:** Lingbo Kong.

**Software:** Yue Yu, Gengyao Li.

**Supervision:** Lingbo Kong.

**Writing – original draft:** Yue Yu, Chaoqun Wang, Zhipeng Xue.

**Writing – review & editing:** Yue Yu, Chaoqun Wang, Zhipeng Xue.

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
