## [Decision Letter · Decision Letter 0]

21 Jan 2022

PONE-D-21-26743System identification and mechanical resonance frequency suppression for servo control used in single gimbal control moment gyroscopePLOS ONE

Dear Dr. Yu,

Thank you for submitting your manuscript to PLOS ONE. After careful consideration, we feel that it has merit but does not fully meet PLOS ONE’s publication criteria as it currently stands. Therefore, we invite you to submit a revised version of the manuscript that addresses the points raised during the review process.

We look forward to receiving your revised manuscript.

Kind regards,

António M. Lopes, PhD

Academic Editor

PLOS ONE

https://journals.plos.org/plosone/s/file?id=ba62/PLOSOne_formatting_sample_title_authors_affiliations.pdf”.

“This work was supported in part by the Jilin Province Scientific and Technological Development Program, grant number 20180201111GX.”

Reviewers' comments:

Reviewer's Responses to Questions

**Comments to the Author**

1. Is the manuscript technically sound, and do the data support the conclusions?

Reviewer #1: Yes

Reviewer #2: Yes

2. Has the statistical analysis been performed appropriately and rigorously? 

Reviewer #1: Yes

Reviewer #2: No

3. Have the authors made all data underlying the findings in their manuscript fully available?

Reviewer #1: Yes

Reviewer #2: Yes

4. Is the manuscript presented in an intelligible fashion and written in standard English?

Reviewer #1: Yes

Reviewer #2: Yes

5. Review Comments to the Author

Reviewer #1: This study focused on the establishment of a model for the SGCMG gimbal servo system, the suppression of mechanical resonance, and the identification of a transfer function. First, the SGCMG gimbal servo system was modelled as the equivalent of a two-mass block model, and the transfer function between the electromagnetic torque and the output angular velocity of the gimbal servo system motor was studied. Second, the typical mechanical resonance frequency in the general servo system was considered. I recommend the article to be accepted for publication in the PLOS ONE after providing following revision.

1. Further strengthen the motivation to write this article in the Abstract section;

2. Please describe in detail the identification and mechanical of problem in paper.

3. Further polish the English writing of the full article;

4. Rewrite the Abstract section, not to detail it, and to focus on the outlines and the task in the paper without going into details or writing letter

5. Please improve the quality of the drawn figures

6. Unify the writing format of all references base on the journal style.

7. More description on Examples should be given,

8. Please also check each reference, there are some typos.

9. The Conclusions section is not satisfactory; it should be further improved. Please emphasize the main novelty of this paper and the significance of the results in the conclusion.

10. Please check writing mathematical formulas in paper.

11. In general, the typeset equations should be regarded as parts of a sentence and treated accordingly with the appropriate grammatical convention and punctuation. More editing for writing is needed. At the end of all equations must be put ”COMMA” or ”POINT” according to the typing rules. Therefore, they need to pre-check all the equations.

12. It is advised to add the following refs related to the work

https://doi.org/10.1016/j.jare.2020.06.018

https://doi.org/10.1016/j.cnsns.2021.105755

https://doi.org/10.1016/j.jocs.2021.101394

Reviewer #2: 1) Considering the current state-of-the-art, what is the novelty of this work?

2) The relevance of the signal processing method (frequency analysis) would be better proven if you add a comparison with those obtained by Fourier and correlation analysis.

3) The experimental result is not clearly presented, with no links between the simulation results and their practical validation.

4) Line 359, “Fig 13 indicates that the fit of the transfer function is excellent”. It is not clear how the authors have concluded the relevance of the fit parameter.

5) The discussion part is poor, need more detail of comprising, you can use some of the related work and mention the references.

6) Line 379, “Therefore, the fitting effect of the transfer function is verified”. The fitting effect is not discussed in the paper. So, the fitting effect should be more detailed.

7) In the introduction part, lines 61, 62, 63, 70, 75, 76, and 79, the word “reference” should be rephrased.

6. PLOS authors have the option to publish the peer review history of their article (what does this mean?). If published, this will include your full peer review and any attached files.

Reviewer #1: No

Reviewer #2: **Yes: **LAHLOUH Ilyas

---

## [Author Response · Author response to Decision Letter 0]

29 Mar 2022

Response to Reviewer 1 Comments

Point 1: Further strengthen the motivation to write this article in the Abstract section;

Response 1: Thank you for pointing this out. We have added a sentence to further strengthen the motivation to write this article in the Abstract.

Point 2: Please describe in detail the identification and mechanical of problem in paper.

Response 2: Thank you for pointing this out. We have added a detailed description of the identification and mechanical of problem in lines 50-56 of the paper.

Point 3: Further polish the English writing of the full article;

Response 3: The manuscript has been edited to improve spelling, grammar and readability.

Point 4: Rewrite the Abstract section, not to detail it, and to focus on the outlines and the task in the paper without going into details or writing letter.

Response 4: Thank you for pointing this out. We have rewritten the abstract. The rewrite does not go into detail but focuses on the outlines and tasks to make the Abstract clearer.

Point 5: Please improve the quality of the drawn figures

Response 5: Thank you for pointing this out. We have improved the quality of the figures.

Point 6: Unify the writing format of all references base on the journal style

Response 6: Thank you for pointing this out. We have unified the writing format of all references base on the PLOS ONE journal style.

Point 7: More description on Examples should be given,

Response 7: We have added more description of the examples in lines 62 to72. In order to explain the principle and procedure of the experiment more clearly, we have added more detailed description in lines 298–302 and lines 332–349. 

Point 8: Please also check each reference, there are some typos.

Response 8: Thank you for pointing this out. We have checked each reference and revised the typos.

Point 9: The Conclusions section is not satisfactory; it should be further improved. Please emphasize the main novelty of this paper and the significance of the results in the conclusion.

Response 9: Thank you for pointing this out. We have emphasized the main novelty of this paper and the significance of the results in the conclusion.

Point 10: Please check writing mathematical formulas in paper.

Response 10: Thank you for pointing this out. We have checked all the mathematical formulas and revised mistakes.

Point 11: In general, the typeset equations should be regarded as parts of a sentence and treated accordingly with the appropriate grammatical convention and punctuation. More editing for writing is needed. At the end of all equations must be put ”COMMA” or ”POINT” according to the typing rules. Therefore, they need to pre-check all the equations.

Response 11: Thank you for pointing this out. Following your comments, we have edited the full manuscript for this. We have checked each of the equations and added the missing comma or full stop as appropriate.

Point 12: It is advised to add the following refs related to the work

https://doi.org/10.1016/j.jare.2020.06.018

https://doi.org/10.1016/j.cnsns.2021.105755

https://doi.org/10.1016/j.jocs.2021.101394

Response 12: Thank you for this suggestion. We have added citations to the three references in the introduction.

Response to Reviewer 2 Comments

Point 1: Considering the current state-of-the-art, what is the novelty of this work?

Response 1: In this paper, orthogonal correlation analysis is applied to servo system identification for the first time. Other scholars previously only identified relevant parameters, making the identified model inaccurate, or use Fourier transform to identify the frequency characteristic curve of the system. The computer is limited by measurement time. Therefore, the Fourier transform can only be carried out in a limited range; this would reduce the accuracy and causes the noise to affect the system identified frequency characteristic curve. In order to verify the superiority of the orthogonal correlation analysis method proposed in this paper, the orthogonal correlation analysis method is compared with the Fourier analysis; it can be observed from the identification results that the noise on the frequency response characteristic curve identified by the orthogonal correlation analysis method is smaller. In addition, this paper also suggests the following to prevent mechanical resonance: increasing the ratio of the motor rotational inertia to the load rotational inertia, increasing the stiffness of the system, and designing a filter; these steps can improve the stability of the servo system. We have emphasized the main novelty of this paper and the significance of the results in the conclusion.

Point 2: The relevance of the signal processing method (frequency analysis) would be better proven if you add a comparison with those obtained by Fourier and correlation analysis.

Response 2: Thank you for pointing this out. We have added the identification result of Fourier analysis in lines 370–383, and compared with frequency analysis at the same time. The experimental results show that there is more noise in the frequency response characteristic curve identified by Fourier analysis. Therefore, the orthogonal correlation analysis method is used to identify the system.

Point 3: The experimental result is not clearly presented, with no links between the simulation results and their practical validation.

Response 3: We have remedied this by highlighting that the main purpose of this paper is to identify the frequency characteristic curve and transfer function of the system through experiment. In order to clearly explain the experimental result, we have added a detailed discussion of the experimental result in lines 391-398.

Point 4: Line 359, “Fig 13 indicates that the fit of the transfer function is excellent”. It is not clear how the authors have concluded the relevance of the fit parameter.

Response 4: Thank you for pointing this out. We have added the error curve between the measured frequency characteristic and the frequency characteristic of the identified transfer function (Fig 15). It can be seen from this that the absolute error of the amplitude-frequency and phase-frequency characteristics are less than 0.5 dB and 1 °, respectively, in the common frequency band. 

Point 5: The discussion part is poor, need more detail of comprising, you can use some of the related work and mention the references.

Response 5: We have added more detailed discussion. In order to explain the principle and procedure of the experiment more clearly, we have added more detailed description in lines 298–302 and lines 332–349. In order to clearly explain the experimental result, we added detailed discussion of the experimental result in lines 370–383 and lines 391-398. We have also used the related work and mention the references in discussion.

Point 6: Line 379, “Therefore, the fitting effect of the transfer function is verified”. The fitting effect is not discussed in the paper. So, the fitting effect should be more detailed.

Response 6: Thank you for pointing this out. We have added more discussed about the fitting effect in lines 391–398.

Point 7: In the introduction part, lines 61, 62, 63, 70, 75, 76, and 79, the word “reference” should be rephrased.

Response 7: Thank you for pointing this out. We have checked the introduction and rephrased the word “reference” in the appropriate lines.

---

## [Decision Letter · Decision Letter 1]

11 Apr 2022

System identification and mechanical resonance frequency suppression for servo control used in single gimbal control moment gyroscope

PONE-D-21-26743R1

Dear Dr. Yu,

We’re pleased to inform you that your manuscript has been judged scientifically suitable for publication and will be formally accepted for publication once it meets all outstanding technical requirements.

Kind regards,

António M. Lopes, PhD

Academic Editor

PLOS ONE

Additional Editor Comments (optional):

Reviewers' comments:

Reviewer's Responses to Questions

**Comments to the Author**

1. If the authors have adequately addressed your comments raised in a previous round of review and you feel that this manuscript is now acceptable for publication, you may indicate that here to bypass the “Comments to the Author” section, enter your conflict of interest statement in the “Confidential to Editor” section, and submit your "Accept" recommendation.

Reviewer #1: All comments have been addressed

Reviewer #2: All comments have been addressed

2. Is the manuscript technically sound, and do the data support the conclusions?

Reviewer #1: Yes

Reviewer #2: Yes

3. Has the statistical analysis been performed appropriately and rigorously? 

Reviewer #1: Yes

Reviewer #2: Yes

4. Have the authors made all data underlying the findings in their manuscript fully available?

Reviewer #1: Yes

Reviewer #2: Yes

5. Is the manuscript presented in an intelligible fashion and written in standard English?

Reviewer #1: No

Reviewer #2: Yes

6. Review Comments to the Author

Reviewer #1: All comments were addressed. The revised paper can be accepted for publication in the PLOS ONE journal.

Reviewer #2: The new version is well prepared end organised and all the responses are very clear.

The novelty is now well detailed, but it would be better if you add it in the introduction part.

7. PLOS authors have the option to publish the peer review history of their article (what does this mean?). If published, this will include your full peer review and any attached files.

Reviewer #1: No

Reviewer #2: No

---

## [Editor Report · Acceptance letter]

9 Aug 2022

PONE-D-21-26743R1 

System identification and mechanical resonance frequency suppression for servo control used in single gimbal control moment gyroscope 

Dear Dr. Yu:

I'm pleased to inform you that your manuscript has been deemed suitable for publication in PLOS ONE. Congratulations! Your manuscript is now with our production department. 

Kind regards, 

on behalf of

Dr. António M. Lopes 

Academic Editor

PLOS ONE